# Molecular Dynamics Investigation of Hyaluronan in Biolubrication

**DOI:** 10.3390/polym14194031

**Published:** 2022-09-26

**Authors:** Masahiro Susaki, Mitsuhiro Matsumoto

**Affiliations:** Graduate School of Engineering, Kyoto University, Kyoto 615-8540, Japan

**Keywords:** biolubrication, biotribology, dynamic viscosity, hyaluronan, hydrophilic polymer, hydration, electrolyte solution, molecular dynamics simulation

## Abstract

Aqueous solution of strongly hydrophilic biopolymers is known to exhibit excellent lubrication properties in biological systems, such as the synovial fluid in human joints. Several mechanisms have been proposed on the biolubrication of joints, such as the boundary lubrication and the fluid exudation lubrication. In these models, mechanical properties of synovial fluid containing biopolymers are essential. To examine the role of such biopolymers in lubrication, a series of molecular dynamics simulations with an all-atom classical force field model were conducted for aqueous solutions of hyaluronan (hyaluronic acid, HA) under constant shear. After equilibrating the system, the Lees-Edwards boundary condition was imposed, with which a steady state of uniform shear flow was realized. Comparison of HA systems with hydrocarbon (pentadecane, PD) solutions of similar mass concentration indicates that the viscosity of HA solutions is slightly larger in general than that of PDs, due to the strong hydration of HA molecules. Effects of added electrolyte (NaCl) were also discussed in terms of hydration. These findings suggest the role of HA in biolubirication as a load-supporting component, with its flexible character and strong hydration structure.

## 1. Introduction

Human joints in healthy states generally move quite smoothly under severe conditions of high load and low relative speed. A dynamic friction coefficient as small as 0.001–0.020 has been reported for human knee joints [1,2]. The joints generally consist of three main components relevant to this good lubrication. (i) Articular cartilage [3], covering the surface of the bone ends, is a composite of collagen fibers, proteoglycan, and hyaluronan (hyaluronic acid, HA). This soft and resilient tissue supports the bones like cushions. (ii) Synovial fluid [4], filling the cartilage gap and the synovial cavity as lubricant, is aqueous solution of several species of biopolymers such as HA and lubricin. This is typical extracellular fluid of human body. (iii) Joint capsule, which encloses the synovial fluid.

The mechanism of joint lubrication has been discussed from various points of view with practical purposes, such as medical treatment of various joint diseases and development of better joint replacements. In the boundary lubrication model [5,6,7,8], it is assumed that the cartilages are sliding each other in the boundary lubrication mode and the biopolymers (proteoglycan aggregates) on the cartilage surface work as “polymer brush”, preventing the direct contact. Another model, i.e., the fluid pressurization mediated lubrication [9,10], suggests more important role of synovial fluid; as the load is imposed on the joint, synovial fluid exudes from the deformed cartilage, lowering the friction.

In either model, mechanical properties of the synovial fluid are essential; it has fairly high viscosity and shows non-Newtonian (shear thinning in most cases) behaviors under large shear, which are very unique compared with other body fluids [11,12,13,14]. It is often argued that these unique characters are brought by high concentration of biopolymers, especially HA [15,16,17]. Recently, due to its biophysically unique characters, much attention has been paid to HA in wide fields of regenerative medicine and bioengineering [18,19,20,21].

Since the typical scale of the joint lubrication phenomena ranges from atomic (nm in space and ns in time) to macroscopic (mm and s) one, multiscale analysis and modeling are indispensable to elucidate the mechanisms [22]. We have reported the multiscale investigation with molecular dynamics (MD) simulation techniques, such as the deformation of cartilage surface under heavy load in μm scale [23] with a Brownian dynamics simulation, the role of polymer brush extruded from the cartilage surface in 10 nm scale [24] with a coarse-grained model, as well as in nm scale [25] with an all-atom model. In this paper, we focus on the role of HA in biolubrication, especially its strong hydrophilicity, and perform a series of MD simulations with an all-atom model. Since the typical HA in synovial fluid has a large molecular weight, all-atom simulations would be a formidable task and coarse-grained models are often adopted [26]. However, our interest in this research exists in how hydrophilic biopolymers (i.e., HA) affect the lubrication dynamics in aqueous solutions with special attention being paid to the local hydration structure, and thus an all-atom model is more suitable to our purpose.

## 2. Systems and Methods

In this study, dynamic behaviors of aqueous HA solution under shear flow were investigated using MD simulations with all-atom models. The models and methods are described in this section. We adopted the LAMMPS package [27] for all simulations. VMD [28] was utilized to visualize the atomic configurations (snapshots).

### 2.1. Molecular Models

The HA molecule in our simulation consists of four units of D-glucuronic acid and N-acetyl-D-glucosamine as shown in Figure 1. The computational resource requirement forced us to limit the size of HA (the number of units n=4) in this study; we are planning to do simulations with much larger *n*.

As for the force field parameters for HA, the Charmm General Force Field (CGenFF) ver. 4.4 [30,31] was adopted, which describes the intra and inter molecular interactions as the sum of Coulombic, Lennard-Jones (LJ), covalent bond, angle, and dihedral terms. The conventional Lorentz-Berthelot combination rule was used for the LJ interactions.

ParamChem [29,32,33] tools were utilized to assign the index and the CGenFF parameters to each atom. The partial charge on each atomic site, which is the most important in hydration of HA molecules, is summarized in Table 1. Note that HA molecules in aqueous solutions are essentially dissociated at O7 (carboxyl group) sites; in our classical mechanical modeling with CGenFF, this is realized as the total charge on two O7 atoms and the neighboring C2 atom being ≃−0.94e (*e*: elementary charge). An appropriate number of sodium ions ${\mathrm{Na}}^{+}$ were added to the system as counter ions to keep the charge neutrality of the system. The LJ parameters for ${\mathrm{Na}}^{+}$ were taken from Ref. [34].

To study the role of hydrophilicity of solute molecules in lubrication, aqueous “solution” of pentadecane (PD, $C_{15}H_{32}$), a typical hydrocarbon, was also investigated for comparison. The CHARMM32 force field [35] was adopted for PD molecules.

For the water, a rigid rotor model with TIP3P parameters [36] was used instead of the more often adopted TIP4P [36] (four-site) model to save the computational time.

### 2.2. Simulation Method and Conditions

Two cases are compared in this study; (i) aqueous solution of HA, and (ii) PD in water. Although hydrocarbons such as PD are essentially insoluble in water, molecular level investigation of their dynamic behaviors would give insight on the role of hydrophilicity in lubrication.

The simulation systems were prepared and examined with the following steps:Step 1:A given number of solute molecules (HA or PD) and an appropriate number of water molecules are randomly placed in a rectangular simulation cell of 10.0 nm × 2.0 nm × 5.0 nm. The number of water molecules was chosen so that the total mass density was about 1.0 g/cm^3^.Step 2:By executing an MD simulation for 0.25 ns, each system was equilibrated with temperature controlled at T=298 K and pressure at P=1 atm using the Nosé-Hoover type thermostat/barostat [37]. Periodic boundary conditions were applied along all directions. An example of atomic configurations of the equilibrated HA solution is shown in Figure 2.Step 3:The main MD simulation with the shear flow was conducted by adopting the Lees-Edwards (LE) boundary condition [38] along *z* direction, which would realize uniform shear flow along *x* direction in the steady state. During the shear flow simulation, the temperature was still controlled to be T=298 K to suppress the viscous heating. The local temperature was evaluated from the mean kinetic energy of atoms after subtracting the local shear flow speed.

Detailed simulation conditions are summarized in Table 2. The Ewald summation method was used for the long-range Coulombic interactions while the short-range LJ interactions were truncated at 10 Å. The number of PD molecules (fourteen) was chosen so that the total mass of solutes is similar for the HA and the PD systems. The mass density of PD system after equilibration at T=298 K and P=1 atm is slightly lower than that of HA system, i.e., the simulation cell of the PD system is more expanded during the equilibration, suggesting that hydrophobic interaction of PD molecules weakens the local hydration.

With the LE boundary condition, we can control the strength of shear flow via the speed difference $V_{x}$ between the top and bottom boundary. Three different values, $V_{x}=$ 100, 200, and 300 m/s, were mainly examined, which correspond to quite large shear rates ($V_{x}/L_{z}=2\times 10^{10}$, $4\times 10^{10}$, $6\times 10^{10}$ s^−1^, respectively) since the cell size along the *z* direction $L_{z}$ is only 5 nm.

## 3. Results and Discussion

### 3.1. Steady State

The solute molecules start to deform when the LE boundary condition is applied to the equilibrated system. Examples of atomic configuration are shown in Figure 3 for typical cases of $V_{x}=200$ m/s. In the HA system, two HA molecules independently diffuse in the cell, with repeated elongation-contraction shape change, while the PD molecules start to aggregate and finally form a single cluster, which deforms under the shear.

To determine when the system reaches the steady state, time evolution of several physical properties (e.g., potential energy and pressure tensor components) were examined. In Figure 4 the total potential energy $E_{p}$ and the pressure component $P_{zz}$ (pressure normal to the shear, macroscopically corresponding to the imposed load pressure) are plotted, which indicate that the steady state seems to be reached at 2–3 ns.

The time average was taken for data in the period of 3 ns ≤ *t* ≤ 5 ns; $\langle E_{p}\rangle$ and $\langle P_{zz}\rangle$ are shown in Figure 5. The shear speed $V_{x}$ does not much affect $P_{zz}$ in HA systems, while $P_{zz}$ of PD systems increases under larger shear. We suppose that large deformation described in the following section and the resulting increase of solvent contact area of PD clusters causes this $\langle P_{zz}\rangle$ rise with $V_{x}$.

### 3.2. Polymer Shape

Before discussing the shape change in shear flow, the equilibrium size should be first mentioned. The spatial size of polymers in solution is generally discussed in terms of the radius of gyration, $R_{g}$. $R_{g}$ of our HA model at equilibrium is 11.1±0.7 Å. Although we found no experimental data for HA of such small weight, this value is well on the extrapolation of experimental results with small-angle X-ray scattering (SAXS) [39], and the model in our simulation seems thus validated.

To investigate the deformation of solute molecules, the instantaneous end-to-end distance $R_{\mathrm{ee}}\left(t\right)$, averaged over molecules, is plotted in Figure 6 (Left). As expected from the snapshots (Figure 3), HA molecules show large fluctuations of elongation and contraction, probably because deformation hardly affects the hydration energy for such hydrophilic polymers. On the contrary, PD molecules show little fluctuations in shape. Assuming that the steady state is reached at 3 ns, the time average is taken for 3–5 ns and the mean value $\langle R_{\mathrm{ee}}\rangle$ with minimum and maximum in this period is shown in Figure 6 (Right). Slight reduction with $V_{x}$ increase is observed for the HA systems; no dependence on $V_{x}$ is seen for the PD cases, suggesting that the shape of PD in the aggregate is not much affected by the aggregate deformation under shear.

### 3.3. Viscosity

The quantity relevant to lubrication behaviors is the shear stress, or the non-diagonal component of stress tensor, $-P_{xz}$. The dynamic viscosity η is defined as the coefficient of its shear rate dependence, assuming a linear response as
(1)$$\langle -P_{xz}\rangle =\eta \frac{V_{x}}{L_{z}}$$

The fluctuating $P_{xz}$ is shown in Figure 7, where no apparent $V_{x}$ dependences are seen in HA as well as PD systems.

Time average of the shear stress for the last 2 ns is plotted in Figure 8; results for pure water cases are also shown, which were separately calculated with 1666 molecules of the same TIP3P model in 5.0 nm × 2.0 nm × 5.0 nm cell (density at equilibrium 0.985 g/cm${}^{3}$). For all cases (HA, PD, and pure water), the data are well fitted to the linear relation, Equation (Equation 1), in spite of the tremendously large shear rate, which means that all these systems behave as Newtonian fluid, probably because essentially no entanglements exist due to the low molecular weight and the low concentration of polymers.

The evaluated η is summarized in Table 3. For pure water, experimental value $0.89\times 10^{-3}$ Pa·s at 25 °C under atmospheric pressure was reported [40,41]; considering the simple water model (TIP3P rigid rotor), the agreement seems reasonable [42]. Experiments for viscosity of human synovial fluid [12] show shear thinning when the shear rate is larger than 0.1 s^−1^ and give η∼ 0.003–0.02 Pa·s at 1000 s^−1^, which is two orders of magnitude larger than our simulation results. The main reason for this discrepancy should be the difference in molecular weight *M*; the typical value of HA in human synovial fluid is $\bar{M}∼10^{6}$[43], which is much larger than M∼1500 in our simulation. Thus entanglements should significantly contribute to the large η and the viscoelastic behavior [44], although the concentration of the HA solution in our simulation is 10–20 times larger than that in vivo [12].

Although similar mass-based amount of solute molecules are included, the viscosity of HA solution is definitely larger than that of PD mixture. This is contrary to our expectation that hydrophilic polymers such as HA play some role for better lubrication through reducing the viscosity.

Discussion about the viscous behavior of dilute solutions of “macromolecules” [45] is conventionally given based on the intrinsic viscosity [η], which is defined as the dilute limit of relative viscosity as
(2)$$\left[\eta \right]\equiv \lim_{c\to 0}\frac{\eta \left(c\right)/\eta_{0}-1}{c}$$
where *c* is the solute concentration and $\eta_{0}$ the solvent viscosity. Since we have not examined the concentration dependence of η yet, [η] cannot be evaluated, but a rough estimation with the finite *c* is given in Table 3, which again indicates the more viscous property of HA than PD. The value [η]∼4 for HA is close to that of small proteins [45] and short DNAs [46,47].

The increase of [η] in macromolecule solutions has been discussed in terms of the molecular shape factor ν and the swollen (or hydrated) volume $V_{\mathrm{hyd}}$ [45,48], as
(3)$$\left[\eta \right]=\nu V_{\mathrm{hyd}}$$

Since the difference in total volume of solute molecules is not very large between the HA and the PD aggregates in this simulation, we suppose that the large fluctuations in shape of HA should contribute to the larger [η] through the factor ν.

### 3.4. Hydration

The swollen volume $V_{\mathrm{hyd}}$ is often discussed in terms of hydration parameter δ, which is defined as
(4)$$\delta =\left(\frac{V_{\mathrm{hyd}}}{V_{\mathrm{anh}}}-1\right)\rho \bar{v}$$
where ρ is the density of solution, $\bar{v}$ the specific volume of the solute molecule, and $V_{\mathrm{anh}}$ the volume of solute molecule in anhydrous states, respectively [49]. Thus δ indicates how much water is bound to the solute molecule in hydrated conditions. Although several theoretical models have been proposed [45,49,50], direct evaluation of $V_{\mathrm{hyd}}$ with molecular simulations is not straightforward because rigorous definition of hydration (or bound water) is hard to give.

Here instead, the standard structural analysis with the radial distribution function (RDF) was conducted for various site-site pairs. In Figure 9, typical RDFs are shown for the HA system with $V_{x}=100$ m/s. Strong binding is seen only between the O7 site of HA [carboxyl group, Figure 1 (Right)] and the H of water molecules. Based on this RDF plot, a water molecule is defined as “bound” when its H atom is within 2.4 Å distance from any of O7 atoms. The mean number of bound water molecules per O7 site is shown in Figure 10, indicating no $V_{x}$ dependence. This definition of “hydration” gives that about six water molecules are strongly bound to each HA molecule, consisting of four units (n=4) of D-glucuronic acid and N-acetyl-D-glucosamine. This value of hydration (6×18.0/1515.3≃0.07 [g/g]) seems much less than typical δ values of 0.1–0.6 g/g for protein molecules [49], suggesting that more loosely bound water molecules are taken into account in the conventional models. It should be pointed out, however, that a significant amount of hydration exists on specific atomic sites with definite electric charge, like carboxyl groups.

### 3.5. Dependence on Salt Concentration

So far, the simulations were conducted for the “salt-free” systems. In the last part, we carried out similar simulations to examine the effects of added salt. For that purpose, a given number of sodium chloride pairs were added (Table 4). Other system parameters and the simulation procedure are the same as described in Section 2.2. The interaction parameters for Cl${}^{-}$ are also taken from Ref. [34].

The time-averaged end-to-end distance $\langle R_{\mathrm{ee}}\rangle$ for the steady state is plotted in Figure 11. The salt concentration does not affect the shape of PD molecules in the aggregate while HA molecules seem to slightly contract with higher salt concentration, suggesting a mechanism similar to the salting-out phenomena.

The mean shear stress $\langle -P_{xz}\rangle$ is plotted as a function of $V_{x}$ in Figure 12, which again indicates that all systems essentially behave as Newtonian fluid under these extremely large shear rates.

The evaluated dynamic viscosity η is summarized in Figure 13, with the no-polymer cases for reference.

The viscosity of aqueous NaCl solution (without polymers) almost linearly increases with the concentration, which reasonably agrees with experiments [51]. The ion-solvent interactions are responsible for this increase [52]. Adding HA or PD to the NaCl solution does not change this tendency, just increasing η by some constant.

Finally RDF between O7 (carboxyl group oxygen) of HA and H of water molecules was plotted in Figure 14 to examine the hydration of HA.

A sharp first peak similar to Figure 9 is observed, from which the number of bound water molecules are evaluated with the same definition as in Section 3.4. As seen in Figure 15, the hydration becomes weaker with the salt concentration increase, suggesting that the added ions (Na^+^ and Cl^−^) become strongly hydrated and partially deprive HA molecules of their bound water. This reduction of hydration is more apparent in larger shear case, probably because the elongation under large shear (Figure 11) increases the solvent-contact area of HA molecules.

## 4. Conclusions

To examine the role of hydrophilic biopolymers in biolubrication processes, a series of molecular dynamics simulations with all-atom models were performed for aqueous solution of hyaluronan (HA) molecules, a typical hydrophilic substance, under uniform shear. In comparison with hydrophobic molecules, pentadecane (PD), it was found that

HA exhibits large shape fluctuations of elongation and contraction during the shear flow.PD molecules aggregate and form a single cluster soon after shear is imposed while the dispersion of HA is maintained during the simulation.In the steady state, both mixtures of HA and PD essentially behave as Newtonian fluid, i.e., the shear stress is proportional to the shear rate. This is probably due to their low molecular weight and no entanglement.The estimated dynamic viscosity η of HA is slightly larger than that of PD, due to the strong hydration of HA molecules.As for the conventional concept of “hydration” in intrinsic viscosity of biopolymers, its significant (if not all) part can be assigned to water molecules bound to the dissociated groups, such as carboxyl ones.The viscosity is enhanced by adding electrolyte (NaCl). Its concentration dependence is similar for HA and PD, supporting the conventional explanation in terms of strong hydration around Na${}^{+}$ and Cl${}^{-}$ ions.

If hydrophilic biopolymers just increase the viscosity, what is their role in biolubrication, such as in the synovial fluid? One of the possible roles is to support the load and maintain the hydrodynamic or mixed lubrication [1,2]. The findings in this paper support this mechanical role of HA, although the molecular weight of simulated HA models is much lower. It is also suggested that flexibility and resilience of HA lead to large deformation and shape fluctuations under shear, affecting the hydration structures.

However, more detailed understanding about the role of hydrophilicity of solute molecules is needed. To examine the entanglement effects (including self-entanglement), it is essential to use polymers with much higher molecular weight, which is also left for our future study. As for the effects of electrolyte, unique characters of divalent ions such as calcium ion Ca${}^{2+}$ have been reported with experimental techniques, such as conformation change with SAXS [53,54], osmotic pressure measurement and dynamic light scattering (DLS) [54], interaction of HA and Langmuir layers with optical microscopy and X-ray reflectivity measurement [55,56]. These experiments reveal that divalent cations replace monovalent ones (e.g., Na${}^{+}$) in the first hydrated shell, leading to large structural change. Molecular simulation with models similar to our work is certainly capable to elucidate the details of these phenomena.

## Figures and Tables

**Figure 1 polymers-14-04031-f001:**
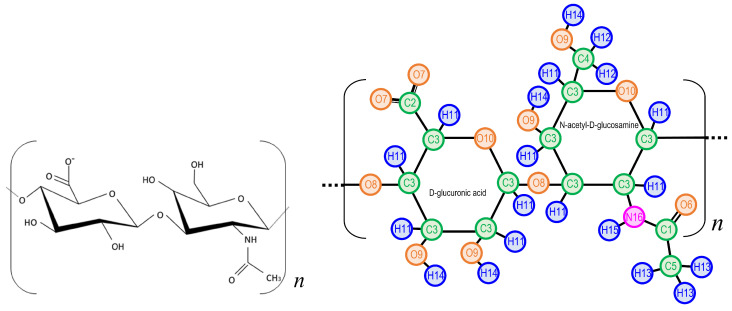
HA model. (**Left**) structural formula, (**Right**) atom type indices assigned with ParamChem [29]. Four unit (n=4) molecules are used in the simulation.

**Figure 2 polymers-14-04031-f002:**
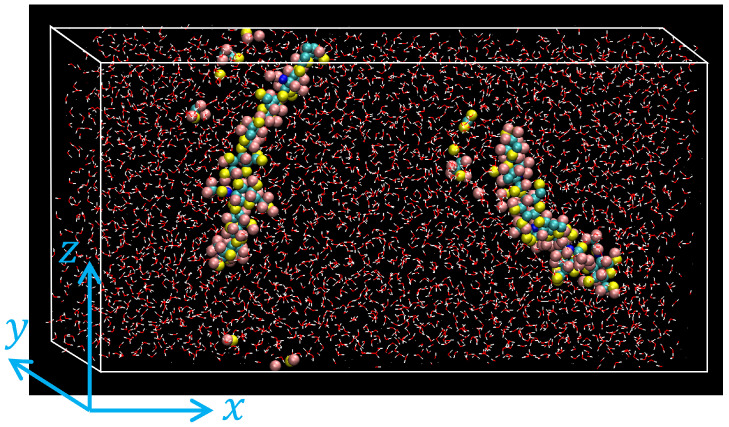
Snapshot of aqueous HA solution after equilibration at T=298 K and P=1 atm. The system contains two HA molecules (shown with van der Waals spheres), which appear to be fragmented due to the periodic boundary conditions. Water molecules are indicated with a wire model.

**Figure 3 polymers-14-04031-f003:**
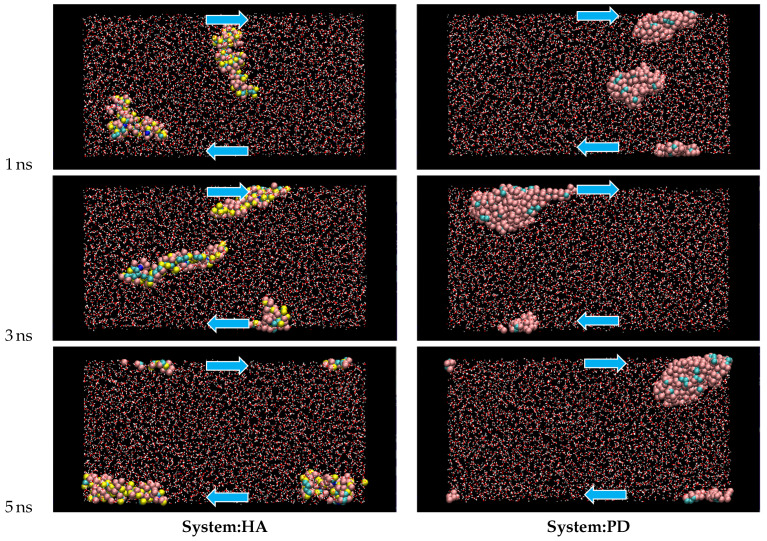
Typical molecular conformation changes during the shear flow simulation; $V_{x}=200$ m/s. The arrows indicate the shear direction. Water molecules are shown with a wire model to clearly indicate the solute molecules. The indicated time is the elapse after imposing the LE condition.

**Figure 4 polymers-14-04031-f004:**
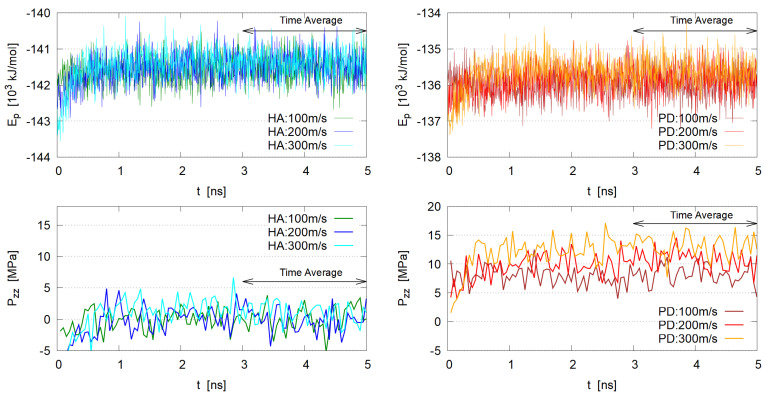
Time evolution of potential energy $E_{p}$ and pressure component $P_{zz}$ normal to the shear direction. (**Left**) HA system, (**Right**) PD system. Since the instantaneous pressure components include large fluctuations, time averaged values in short period (50 ps) are plotted for $P_{zz}$.

**Figure 5 polymers-14-04031-f005:**
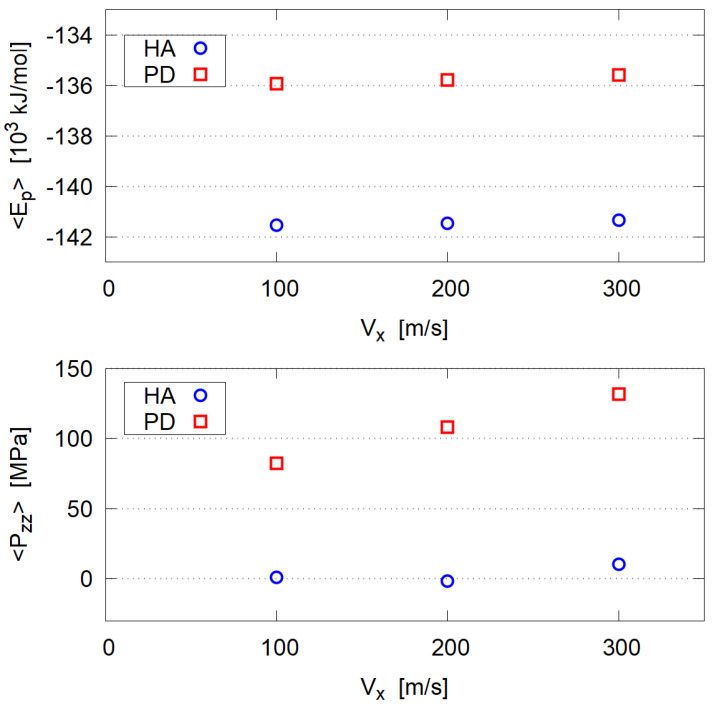
Averaged potential energy and pressure; the time average is taken for period 3 ns ≤ *t* ≤ 5 ns.

**Figure 6 polymers-14-04031-f006:**
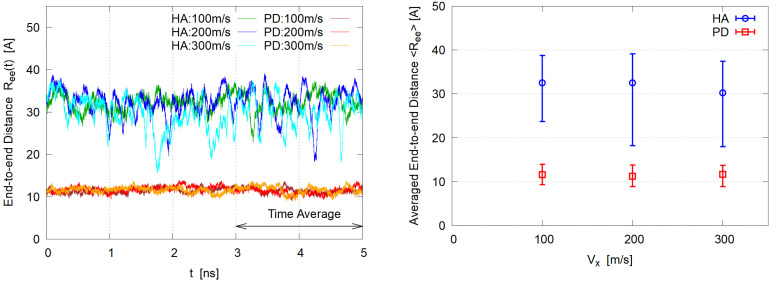
End-to-end distance. (**Left**) time evolution during the simulation, (**Right**) time average with minimum and maximum values for the period 3 ns ≤ *t* ≤ 5 ns.

**Figure 7 polymers-14-04031-f007:**
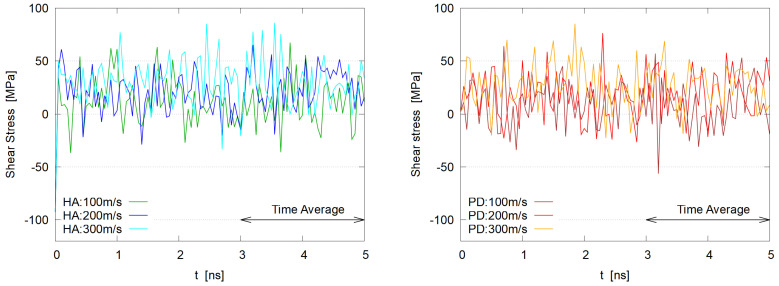
Time evolution of shear stress, $-P_{xz}$ for HA (**left**) and PD (**right**) systems; short period (50 ps) time average is taken, similar to Figure 4.

**Figure 8 polymers-14-04031-f008:**
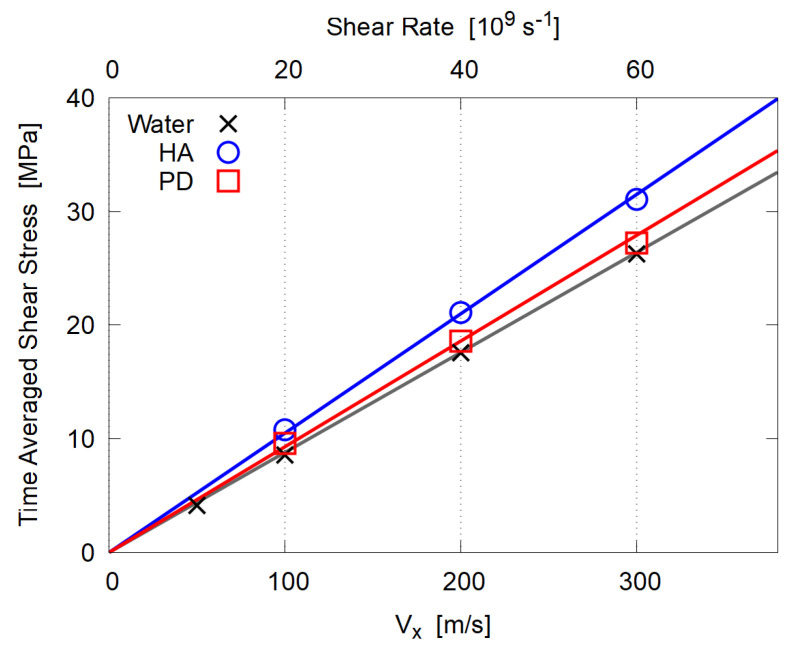
Time averaged shear stress $\langle -P_{xz}\rangle$ plotted against the shear speed $V_{x}$. The lines indicate the least square fitting to the linear relation, Equation (Equation 1), to evaluate the dynamic viscosity η.

**Figure 9 polymers-14-04031-f009:**
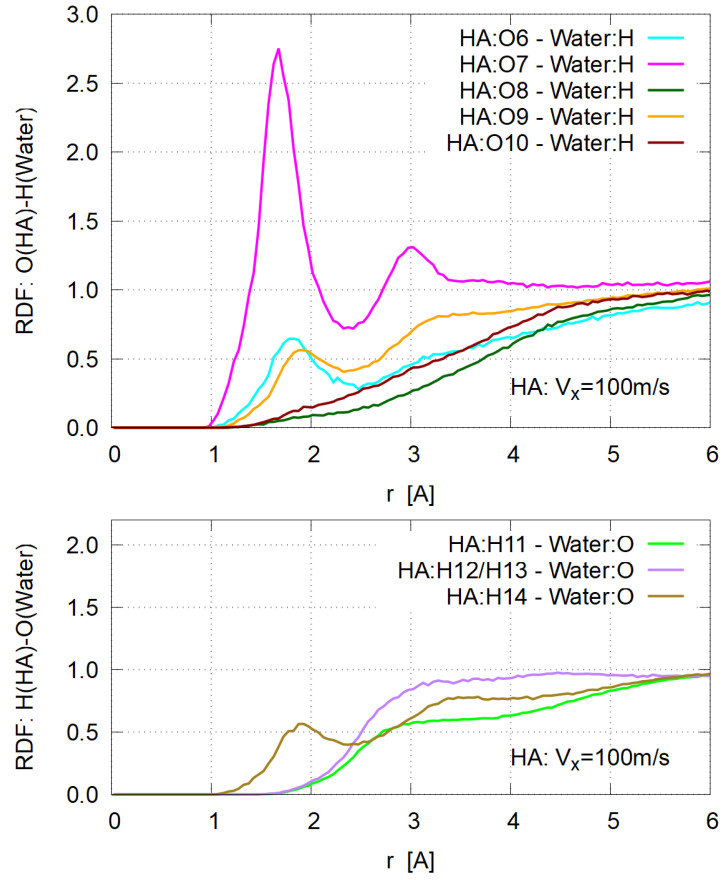
Atom-atom radial distribution functions for HA system with $V_{x}=$ 100 m/s; (**Top**) Between oxygen atoms of HA and hydrogen of water, (**Bottom**) between hydrogen atoms of HA and oxygen of water.

**Figure 10 polymers-14-04031-f010:**
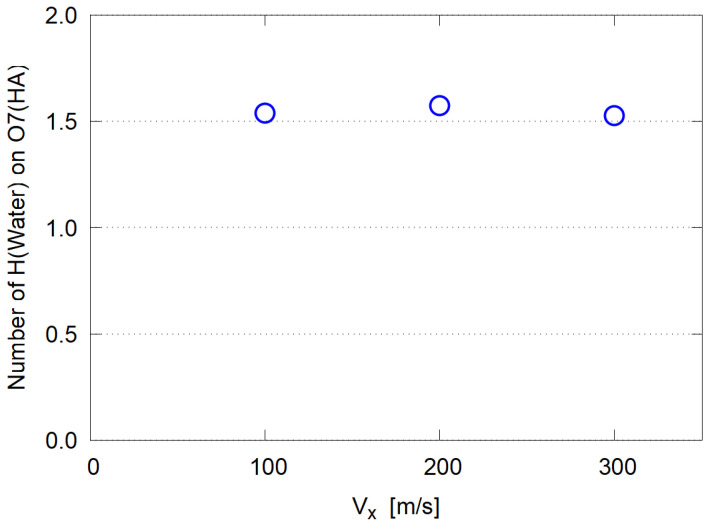
Average number of water molecules bound to HA:O7 sites.

**Figure 11 polymers-14-04031-f011:**
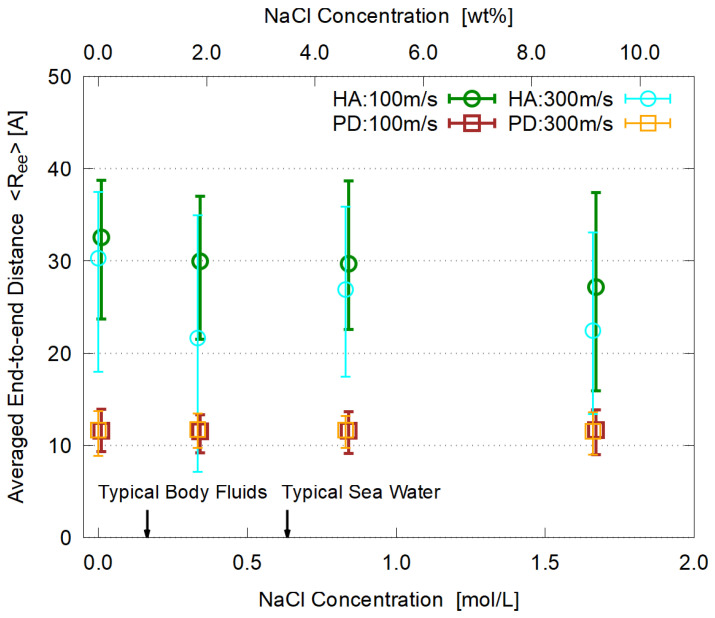
Salt concentration dependence of the averaged end-to-end distance with minimum and maximum, time-averaged for 3 ns ≤t≤ 5 ns.

**Figure 12 polymers-14-04031-f012:**
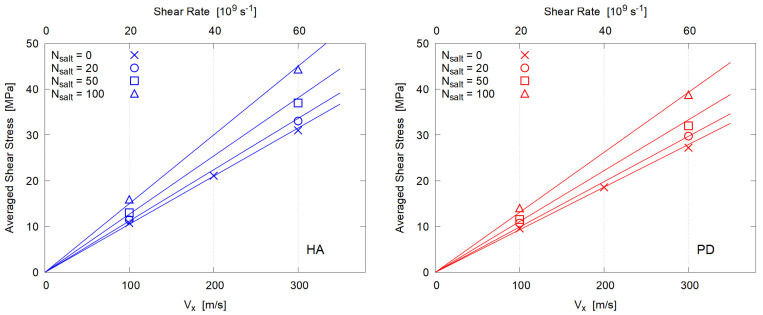
Shear rate dependence of the averaged shear stress.

**Figure 13 polymers-14-04031-f013:**
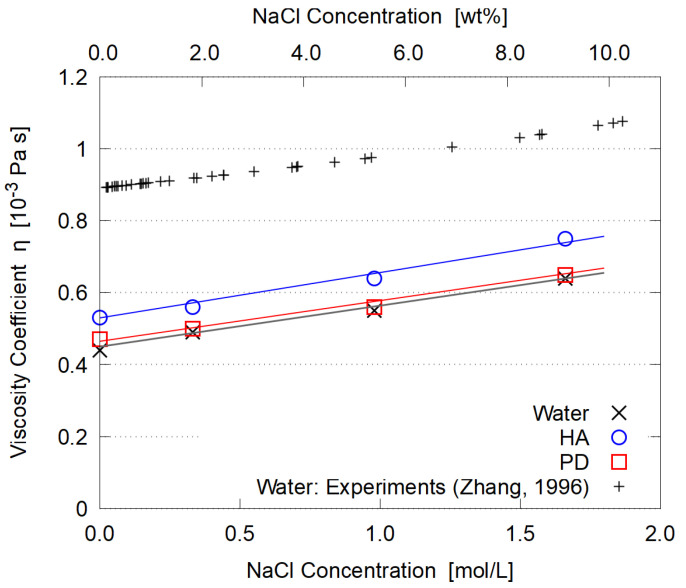
Salt concentration dependence of dynamic viscosity η. Experimental data for NaCl solutions (without polymers) are taken from Ref. [51]. Lines are drawn as a guide for the eye.

**Figure 14 polymers-14-04031-f014:**
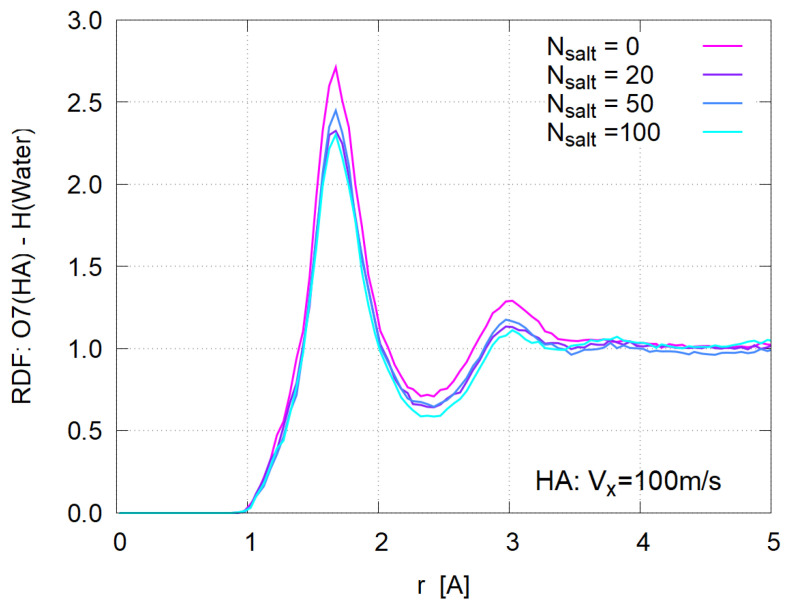
Atom-atom radial distribution function for HA system for $V_{x}=$ 100 m/s case.

**Figure 15 polymers-14-04031-f015:**
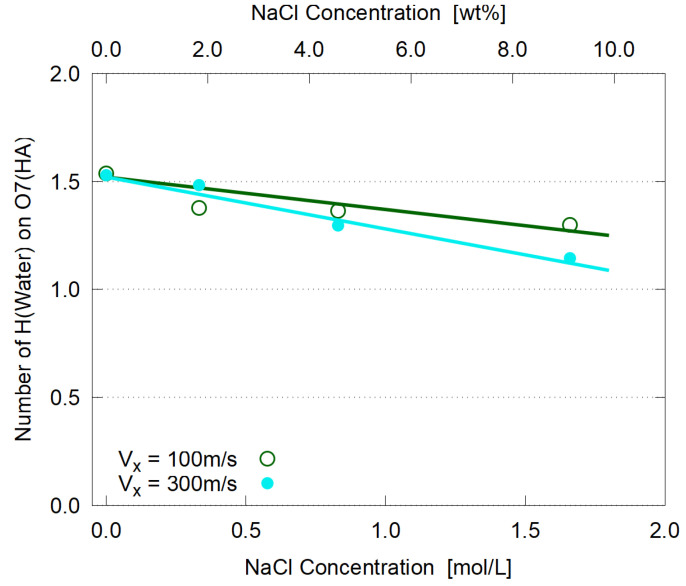
Coordination number, or the number of bound water molecules, on each O7 site of HA molecules; the lines are guides to the eye.

**Table 1 polymers-14-04031-t001:** Partial charge assigned to each atom type of HA molecule by CGenFF ver. 4.4 [32]. The exact value for atom type with (*) is determined by more detailed local structure.

Element	Atom Type	Charge [*e*]	Element	Atom Type	Charge [*e*]
Carbon	C1	0.545	Hydrogen	H11	0.090
	C2	0.579		H12	0.090
	C3 (*)	−0.046∼0.291		H13	0.090
	C4	0.100		H14 (*)	0.244∼0.419
	C5	−0.269		H15	0.090
Oxygen	O6	−0.507	Nitrogen	N16	−0.405
	O7	−0.760			
	O8 (*)	−0.313∼−0.307			
	O9 (*)	−0.670∼−0.647			
	O10 (*)	−0.320∼−0.284			

**Table 2 polymers-14-04031-t002:** System parameters for the simulation.

		System:HA	System:PD
Solute	Molecular formula	${\left(C_{14}H_{20}NO_{11}^{-}\right)}_{4}$	$C_{15}H_{32}$
	Number of atoms	187	47
	Molecular weight	1515.3	212.4
	Number of molecules	2	14
Counter ion	Number of ${\mathrm{Na}}^{+}$	8	–
Solvent	Number of $H_{2}O$	3332	3332
	Total number of atoms	10,376	10,654
	Initial cell size [Å]	100.0 × 20.0 × 50.0	
	Cell size after equilibration [Å]	101.6 × 20.3 × 50.0	103.4 × 20.6 × 50.0
	Density after equilibration [g/cm^3^]	1.02	0.98
	Time step [fs]	0.5	

**Table 3 polymers-14-04031-t003:** Evaluated dynamic viscosity η and rough estimation of intrinsic viscosity [η].

	Pure Water	System:HA	System:PD
η [$10^{-3}$ Pa·s]	0.44	0.53	0.47
Concentration *c* [g/cm^3^]	−	0.0490	0.0464
$\left[\eta \right]≃\frac{\eta /\eta_{0}-1}{c}$ [cm^3^/g]	−	4.2	1.5

**Table 4 polymers-14-04031-t004:** Simulation conditions for salt effect investigation.

Number of Added NaCl Pairs	Concentration	
	**[mol/L]**	**[wt%]**
0	0	0
20	0.321	1.82
50	0.802	4.42
100	1.588	8.55

## Data Availability

The related data are available on request from the corresponding author.

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
