# Peer review of "Molecular Dynamics Investigation of Hyaluronan in Biolubrication"

_polymers, 2022, doi:10.3390/polym14194031_

Round 1

Reviewer 1 Report

The authors have done very good work! English is good, Design of the experiment is good, and the Conclusions are supported by the work done. 

I would recommend an acceptance for this manuscript.

Author Response

We appreciate your time and effort in reviewing our manuscript. Several misspellings and grammatical errors have been corrected in the revised manuscript.

Reviewer 2 Report

The manuscript submitted by Masahiro Susaki and Mitsuhiro Matsumoto reported on the “Molecular Dynamics Investigation of Hyaluronan in Biolubrication” In this work, hyaluronic acid and pentadecane were prepared and tested. Viscosity test, shear flow, salt concentration dependence and calculation analysis were used to analyze those materials. The paper looks very good and detailed experiments were listed. I think only minor modification is needed before publication.

Please slightly rephrase your abstract. After the first sentence talking about the background, the authors need at least one sentence to address what is the topic of this paper. And then taking about “To examine the role of….”

It is kind of weird to put a figure in the section of conclusion. Please move it to the discussion part and let the conclusion part be just conclusion.

This paper was well-prepared and many necessary citations were listed. I believe it can be published after minor modification

Author Response

We much appreciate your comments and suggestions.

(1) “Abstract” was revised, by adding some description on the importance of HA properties in biolubrication, and how our findings contribute to the understanding of biolubrication mechanism.

(2) We agree that the last figure should appear before “Conclusion”, but its placement is (almost) automatically determined by the LaTeX system. We did our best in the revised manuscript, and we will check it in the proof-reading process.

Reviewer 3 Report

Manuscript “Molecular Dynamics Investigation of Hyaluronan in Biolubrication” by Matsumoto et. al. (polymers-1915515) is well interesting example of molecular dynamic application in getting understanding of biolubrication. It is hot topic of current multidiscipline research. Authors modelled behavior of key component of lubrication process i. e., biopolymer Hualoronan. Structural properties of Hualoronan solutions at steady state flow were obtained and compared with hydrocarbon (pentadecan). On my mind authors should considered few additional points which could improve manuscript.

1.    In abstract and conclusions outlook points are missing, how presented calculations can contribute to biolubrication phenomen.

2.    Authors use quite a lot of abbreviation, may be better to prepare separate list of abbreviations.

3.    It is clear that modelling results should be checked and used by other studies. Is it possible to model radius of gyration (Rg) which can be easily check by scattering methods?

4.    It is known that Ca ions play specific role on structures in biolubrication (10.1039/c9sm01066a), is it possible to extend modelling for divalent ions? Or just comment this?

Author Response

We very much appreciate your comments and suggestions. We have add more data, discussion, and relevant references in the revised manuscript.

(1) Description was added in “Abstract” and “Conclusion” about the contribution of our findings to the research field of biolubrication.

(2) It does not seem a usual custom in this journal, Polymers, to include “nomenclature” or “abbreviation list”, but we prepared it at the last part. We ask the editorial office about the style.

(3) We are grateful for the suggestion about the radius of gyration (Rg) data. Although the HA molecule in our simulation is much smaller than those used in experimental investigation, we have confirmed that <Rg> in our simulation seems well on the extrapolation of SAXS data [Mizukoshi et al., 1998, added in the revised manuscript]. Description has been added at the beginning of Sec. 3.2.

(4) We much appreciate the comment on the effects of divalent ions (Ca2+); we have not noticed such drastic difference of monovalent/divalent cations. We have not completed the reference survey on this effect, but four seemingly-relevant references have been added, including your suggested one. In principle, a similar simulation technique (i.e., a full-atom MD simulations with classical force field models) should be applicable to investigate the detail, which is left for our future work. Still, we totally agree that the difference of monovalent/divalent ions are important also in practical situations, and some description has been added at the last part of “Conclusion”.